# Dysfunctional Autophagy, Proteostasis, and Mitochondria as a Prelude to Age-Related Macular Degeneration

**DOI:** 10.3390/ijms24108763

**Published:** 2023-05-15

**Authors:** Raji Rajesh Lenin, Yi Hui Koh, Zheting Zhang, Yan Zhuang Yeo, Bhav Harshad Parikh, Ivan Seah, Wendy Wong, Xinyi Su

**Affiliations:** 1Department of Ophthalmology, Yong Loo Lin School of Medicine, National University of Singapore (NUS), 1E Kent Ridge Road, NUHS Tower Block Level 7, Singapore 119228, Singapore; 2Department of Medical Research, SRM Institute of Science and Technology, Kattankulathur 603203, Tamil Nadu, India; 3Lee Kong Chian School of Medicine, Nanyang Technological University (NTU), 11 Mandalay Road, Experimental Medicine Building, Singapore 308232, Singapore; 4Institute of Molecular and Cell Biology (IMCB), Agency for Science, Technology, and Research (A*STAR), 61 Biopolis Drive, Proteos, Singapore 138673, Singapore; 5Department of Ophthalmology, National University Hospital (NUH), 1E Kent Ridge Road, NUHS Tower Block Level 7, Singapore 119228, Singapore; 6Singapore Eye Research Institute (SERI), The Academia, 20 College Road, Level 6 Discovery Tower, Singapore 169856, Singapore

**Keywords:** mitochondrial dysfunction, age-related macular degeneration, retinal pigment epithelium, aging, clinical trials, autophagy

## Abstract

Retinal pigment epithelial (RPE) cell dysfunction is a key driving force of AMD. RPE cells form a metabolic interface between photoreceptors and choriocapillaris, performing essential functions for retinal homeostasis. Through their multiple functions, RPE cells are constantly exposed to oxidative stress, which leads to the accumulation of damaged proteins, lipids, nucleic acids, and cellular organelles, including mitochondria. As miniature chemical engines of the cell, self-replicating mitochondria are heavily implicated in the aging process through a variety of mechanisms. In the eye, mitochondrial dysfunction is strongly associated with several diseases, including age-related macular degeneration (AMD), which is a leading cause of irreversible vision loss in millions of people globally. Aged mitochondria exhibit decreased rates of oxidative phosphorylation, increased reactive oxygen species (ROS) generation, and increased numbers of mitochondrial DNA mutations. Mitochondrial bioenergetics and autophagy decline during aging because of insufficient free radical scavenger systems, the impairment of DNA repair mechanisms, and reductions in mitochondrial turnover. Recent research has uncovered a much more complex role of mitochondrial function and cytosolic protein translation and proteostasis in AMD pathogenesis. The coupling of autophagy and mitochondrial apoptosis modulates the proteostasis and aging processes. This review aims to summarise and provide a perspective on (i) the current evidence of autophagy, proteostasis, and mitochondrial dysfunction in dry AMD; (ii) current in vitro and in vivo disease models relevant to assessing mitochondrial dysfunction in AMD, and their utility in drug screening; and (iii) ongoing clinical trials targeting mitochondrial dysfunction for AMD therapeutics.

## 1. Introduction

Age-related macular degeneration (AMD) is the leading cause of irreversible blindness in developed countries [1]. Central vision loss occurs in end-stage AMD, a disease that has been rising in prevalence in recent years amongst aging populations. Dry AMD accounts for 85% to 90% of AMD cases, hence a comprehensive understanding of the global dry AMD burden is needed [2]. With an aging global population, AMD will become the leading cause of blindness in developed countries, with an estimated 288 million patients by 2040 [3]. This necessitates greater efforts to reduce the disease impact of AMD.

Early AMD is characterised by drusen and pigmentary abnormalities, whilst vitelliform lesions are observed in late-stage AMD, where the defining lesions are macular neovascularisation (MNV) [4] and geographic atrophy (GA) [5]. Late-stage dry AMD is characterised by discrete areas of RPE loss and the degeneration of overlying retinal photoreceptor cells. Drusen, extracellular deposits of lipids, also accumulate between the basal lamina layer of retinal pigment epithelial (RPE) cells and the inner collagenous layer of Bruch’s membrane. Meanwhile, wet AMD is characterised by the penetration of choroidal capillaries through Bruch’s membrane, resulting in RPE damage and haemorrhage [6]. Despite 85% of AMD cases being dry AMD [2], only one FDA-approved treatment option exists for dry AMD.

Syfovre^TM^ was recently approved by the FDA for the treatment of dry AMD. It is a pegcetacoplan injection which inhibits C3 and regulates the activation of complement pathways [7]. It aims to slow the progression of dry AMD, with continual treatment reported to lead to increasing effects. Nevertheless, it should be noted that Syfovre^TM^ requires monthly to bi-monthly injections, which may lead to side effects in the long term. Neovascularisation has been reported in some participants as well [7]. Hence, there is still a need to discover more treatment options.

AMD is a complex disease with several pathways plausibly involved in its pathogenesis. This poses several therapeutic challenges. The contemporary management of early AMD depends primarily on observation, lifestyle changes, frequent follow-up evaluations, the early recognition of visual deterioration, and CNV detection. Several therapeutic avenues to reduce the rate of disease progression are being investigated, including (1) drugs/supplements with anti-oxidative properties, (2) inhibitors of the complement cascade, (3) neuroprotective agents, (4) visual cycle inhibitors, (5) gene therapy, and (6) and cell-based therapies.

Mitochondrial ROS and oxidative damage are significant contributors to mitochondrial dysfunction and ER stress, which increase with aging and aging-related diseases [8,9]. Impaired mitochondrial function leads to a decline in the autophagic capacity and the induction of inflammation and apoptosis in human RPE cells, contributing to AMD [10]. Changes in the mitochondria of RPE, including mtDNA deletion and mutation, decreased ATP production, mitochondrial fission/fusion imbalance, decreased mitochondrial biogenesis, and mitophagy, have commonly been observed during RPE aging and degeneration. Based on these central roles of mitochondria in AMD, strategies targeting mitochondrial homeostasis have great potential. Recent advances highlight the importance of mitochondrial homeostasis in disease mechanisms; however, several outstanding questions and key issues remain to be investigated.

This review aims to summarise and provide a perspective on (i) the current evidence of mitochondrial dysfunction in dry AMD; (ii) current in-vitro and in-vivo disease models relevant to assessing mitochondrial dysfunction in AMD, and their role in drug screening; and (iii) ongoing clinical trials looking at targeting mitochondrial dysfunction for AMD therapeutics.

## 2. Dysfunctional Autophagy, Proteostasis, and Mitochondria in AMD Pathogenesis

Aging results in the impairment of proteostasis, which is typically carried out via the autophagic and proteasomal pathways. Autophagy is responsible for the degradation of damaged organelles or damaged proteins that are unable to be cleared by other processes [11]. RPE cells phagocytose photoreceptor outer segments (POS). However, with aging, dysfunctions in lysosomal processing [12] affect POS breakdown. The oxidation of lipid deposits forms indigestible lipofuscin, which builds up in the RPE layer [13]. These waste products are exocytosed and deposited between the RPE layer and Bruch’s membrane to form drusen, a key pathologic hallmark in AMD patients [9,14]. The resulting thickening of Bruch’s membrane impedes trans-epithelial transportation of nutrients, ions, and waste between the choriocapillaris and RPE layer, ultimately leading to RPE loss—a characteristic of geographic atrophy in dry AMD [14,15].

The proteasomal pathway, which degrades damaged or unneeded proteins, is inhibited during aging [15]. This could be attributed to age-related changes in gene expression pathways [16]. Lipofuscin aggregates, which form during aging, were also found to inhibit the proteasomal pathway by competitively binding to proteolytic enzymes [17].

Aging, as per the mitochondrial–lysosomal axis theory, is a multifaceted process, and is also associated with the accumulation of enlarged, dysfunctional mitochondria [13]. In RPE cells, accumulated mitochondrial DNA (mtDNA) mutations lead to mtDNA heteroplasmy and changes in mitochondrial structure [11]. Mitochondrial fusion/fission balance is disrupted, decreasing mitophagy (a subset of autophagy pertaining to the degradation of mitochondria) [18]. This results in accumulation of dysfunctional mitochondria in the RPE cells, leading to the decline in mitochondria function and bioenergetics.

MtDNA damage in RPE cells has been observed to correlate with AMD severity in patients, who also exhibited more mtDNA damage compared with age-matched controls [19]. Moreover, reduced mitochondrial function was also reported in AMD patients [20]. This substantiates the redox theory of aging, which states that an overwhelming amount of oxidative stress due to mitochondrial dysfunction disrupts redox signalling, causing macromolecular damage, and ultimately, cell death [21].

Therefore, AMD is a result of dysfunctional autophagy, proteostasis, and mitochondria. These processes are essential to ensure that the high metabolic demands of the RPE cells are fulfilled, and to ensure that the RPE layer is able to fully support photoreceptors. When one of these processes starts to fail and accumulates adverse effects, due to aging or genetic mutations, the onset of AMD occurs.

## 3. Modelling Dry AMD In Vitro

Difficulties in developing novel therapeutics for AMD include identifying a single effective drug for different stages of AMD, and a lack of efficacy in current experimental drugs in reducing the rate of progression. Developing in vitro disease models will be helpful in assessing novel therapeutics for AMD. In vitro models should aim to recapitulate features of AMD phenotypes such as lipid deposition, cell atrophy, and more, depending on their study objectives. Stem-cell-derived two-dimensional (2D) culture models are widely used because of the high reproducibility, ease of use, cost effectiveness, and applicability for long-term experiments and large-scale studies [22]. However, some limitations include their inability to represent tissues in vivo [23]. Nevertheless, great efforts have been made to generate suitable in vitro models to test potential treatments and minimise the use of animals. To simulate conditions of aging and degeneration in the retina, many studies have applied various stressors and specialised conditions to cell culture systems. The AMD Disease Models task group of the Ryan Initiative for Macular Research (RIMR) provided a summary of several in vitro RPE models currently in use [24]. The paper extensively covers various models of in vitro AMD that are generated from various RPE cell lines.

### 3.1. Modelling by Oxidative Stress Induction in RPE Cells

Broadly, chemical inducers of oxidative stress have shown variable and inconsistent phenotypic changes. Furthermore, light-induced studies mentioned here also warrant further research. Phenotypic changes exhibited in oxidative stress models include increased (i) cytotoxicity; (ii) intracellular ROS production; (iii) expression of pro-inflammatory cytokines; (iv) mitochondrial dysfunction (e.g., mtDNA damage and decreased mitochondrial membrane potential; and (v) disruption of junctional integrity. In this section, we discuss the various chemical and light-induced methods that have been used to induce oxidative stress in RPE cells.

#### 3.1.1. Chemical-Induced Oxidative Stress Models

Common agents used to induce oxidative stress in RPE cells for in vitro AMD models include hydrogen peroxide (H_2_O_2_), 4-hydroxynonenal (HNE), tert-butyl hydroperoxide (TBHP), blue light, and ultraviolet light. However, the concentration used has varied from study to study; for instance, typical concentrations of H_2_O_2_ range from 100 μM to 1 nM [25,26].

H_2_O_2_ is the single most used stressor to model AMD in vitro. H_2_O_2_ occurs in normal metabolism in mammalian cells and is a key metabolite in oxidative stress. Excessive H_2_O_2_, beyond physiological levels, leads to oxidative damage in various cell types, including retinal cells [27]. It has repeatedly been found across multiple studies that H_2_O_2_ dose-dependently inhibits RPE cell survival [28,29].

TBHP is another commonly used chemical for the exogenous induction of oxidative stress in cells and tissues. Compared to H_2_O_2_, TBHP is less reactive and more soluble in organic solvents. However, there appears to be significant variation in protocols using TBHP. Variations include the concentration of TBHP and the incubation period.

With published studies using different methods and parameters to induce oxidative stress as mentioned above, this brings forth broad uncertainties in replicating an in vitro AMD model. In addition, this prompts us to think more deeply about the reliability of the data obtained from different models used in the studies, hence emphasising the importance of future research performing in-depth literature reviews and in vitro model optimisation.

#### 3.1.2. Light-Induced Oxidative Stress Models

The eye is exposed to a broad spectrum of light daily. The cornea and lens absorb ultraviolet (UV) light below 400 nm, whereas the visible light component (380–780 nm) of optical radiation can penetrate the retina. In particular, blue light (400–500 nm) is known to cause retinal damage due to its relatively high energy [30,31]. A2E is a prominent constituent of lipofuscin that enhances cellular sensitivity to light radiation. This photosensitisation leads to oxidative stress and cell death [32]. Prior studies have shown that RPE cells fed with A2E for 2 h led to intracellular accumulation of A2E. Subsequently, these RPE cells were exposed to blue light, and it was found that they had undergone photosensitisation. Increased toxicity, oxidative stress, and cell death were observed [33].

Several studies have shown that ultraviolet B (UVB) light induces direct DNA damage and oxidative stress in RPE cells by increasing ROS and dysregulating endogenous antioxidants. Furthermore, UVB irradiation also triggers inflammation and cell apoptosis via various pathways. Exposure to UVB also alters the autophagy, phagocytosis, and permeability of RPE cells [34].

### 3.2. Modelling Using Patient-Derived Cell Lines with Risk Alleles to Demonstrate AMD Phenotypes In Vitro

Sub-RPE deposits such as drusen represent a key pathology in dry AMD. In vitro models utilising patient-derived cell lines with risk alleles have been shown to be able to induce formation of sub-RPE deposits [35]. Human iPSC-RPE cells derived from patients with dominant macular degenerations (Sorsby fundus dystrophy and Doyne honeycomb retinal dystrophy) have been used to model in vitro sub-RPE deposit formation [36]. Notably, cultures from patients with macular degeneration developed a greater number of deposits with a different composition when compared with normal-age matched individuals, highlighting the potential value of using these model systems to elucidate disease-specific mechanisms that lead to drusen formation. Sharma et al. extensively characterised and demonstrated AMD phenotypes in patient-derived RPE cells with CFH risk allele after exposure to complement-competent human serum [37]. However, it has yet to be shown if mitochondria dysfunction occurs in that model. Another study by Saini et al. also uses RPE reprogrammed from AMD patients, whereby nicotinamide alone was shown to ameliorate AMD disease-related phenotypes by inhibiting drusen proteins and inflammatory and complement factors while upregulating nucleosome-, ribosome-, and chromatin-modifying genes [38].

### 3.3. Summary of In Vitro Disease Modelling

Some of the criticisms around the in vitro modelling of RPE cells would be that the in vitro phenotype is not indicative of the in vivo phenotype, and only acute doses of stress may be used, which might not be entirely representative of the in vivo conditions of progressive aging. Aging is a chronic process with pathological changes developing gradually over a prolonged period [39]. Therefore, future in vitro modelling of AMD could further explore models of chronic stress and aging, in tandem with the aforementioned multicellular models.

Importantly, AMD is a complex disease incompletely modelled by both existing in vivo and in vitro models. No animal models developed thus far have been able to fully recapitulate human AMD [40,41]. Similarly, in vitro models only replicate certain aspects of AMD and fail to capture all the associated characteristics [42]. Despite this and other limitations, both model types also have distinct advantages. In vitro models are widely accessible and enable controlled experimentation, while disease progression in animal models more closely resembles that of human AMD, emphasising its vital clinical significance in AMD research.

## 4. In Vivo Models for Investigating Dry AMD Phenotypes

### 4.1. NRF2/PGC-1*α* and RB1CC1-Deficient Mouse Models

#### 4.1.1. NRF2

Nuclear factor erythroid 2-related factor 2 (NRF2/NFE2L2) is a master antioxidant transcription factor. One of its many roles include protecting cells from oxidative damage. NRF2^−/−^ mice present with drusen-like deposits, RPE atrophy, RPE pigmentary changes, increased autofluorescence, complement deposition, and reduced electroretinograms (ERGs). Drusen formation was observed in the mice from 8 months onwards. At 12 months, extensive vacuolation, pigmentary changes, and loss of RPE are observed. Bruch’s membrane was also significantly thickened. The study also found swollen mitochondria, readily detectable autophagosomes, autolysosomes and increased lipofuscin, suggesting hindered autophagy in this model [43].

#### 4.1.2. PGC-1

The positive regulator of PGC-1 (peroxisome proliferator-activated receptor gamma coactivator-1) is a master regulator of mitochondrial biogenesis and oxidative metabolism. Zhang et al. generated a PGC-1*α*^+/−^ mouse model to study the pathogenesis of AMD. PGC-1*α*^+/−^ mice expressed lower levels of PGC-1*α* and were fed a high-fat diet for 4 months. The mice displayed drusen and lipofuscin accumulation, elevated ROS levels, decreased autophagy flux, and increased inflammation, in addition to clear RPE and photoreceptor cell degeneration [44]. This mice model was used to demonstrate that PGC-1*α* is a key player for the regulation of autophagy, playing a role in preventing oxidative damage.

#### 4.1.3. NRF 2/PGC–1*α* Double-KO

Felszeghy et al. [45] further explored the autophagy-regulated functions of both NRF2 and PGC-1*α* pathways in the development of dry AMD. They established and characterised an NRF2/PGC-1*α* double-KO (dKO) mouse model to investigate the role of autophagy clearance in regulating the antioxidant response. NRF2/PGC-1*α* dKO mice developed severe AMD with the accumulation of oxidative stress markers and damaged mitochondria. The severity in NRF2/PGC-1*α* dKO mice being higher than that in the single KO (NRF2 or PGC1) mice indicates that both the genes synergistically contribute to the control of stress regulation. RPE cells from dKO mice exhibited larger autolysosomes and a higher ratio of damaged mitochondria than RPE cells from WT mice, as visualised by transmission electron microscopy [45]. The study not only highlighted the significant role of intracellular degradation systems, including autophagy and the UPS in reducing oxidative stress, but also revealed potential crosstalk between the NRF2 and PGC-1*α* pathways. Twelve-month-old NRF2/PGC-1α-deficient mice are associated with drusen-like deposition, RPE degeneration, pigmentary changes, thickened Bruch’s membrane, and photoreceptor loss, resulting in decreased ERGs and immune cell activation [45]. These mice demonstrate thinning and disorganisation of the outer nuclear layer (ONL); Iba-1 levels were elevated, a marker of macrophage/microglial activation [46]. Additionally, lipofuscin-like granules were observed with larger autolysosomes and fewer mitochondria, substantiating the theory of aging. A separate study revealed that in this mouse model, RPE cells initiated mitophagy but autophagic flux was disturbed, impeding mitochondrial clearance [47].

#### 4.1.4. RB1CC1

RB1CC1 knockout mice were generated to study the relation between reduced autophagy and RPE dysfunction during AMD. The gene encoding RB1CC1/FIP200 (RB1-inducible coiled-coil 1), a protein essential for the induction of autophagy, was selectively knocked out in the RPE layer by crossing Best1-Cre mice with mice in which the Rb1cc1 gene was flanked with Lox-P sites (Rb1cc1flox/flox). RB1CC1 knockout mice presented with geographic atrophy, drusen, increased autofluorescence, complement deposition, loss of photoreceptors, reduced ERGs, and immune cell accumulation. At 4 months, small, white-yellowish structures resembling drusen, patches of RPE atrophy, and RPE hyperpigmentation at the border of these patches were observed. At 8 months, it was observed that AMD phenotypes worsened in the RB1CC1 knockout mice, and Iba-1-positive cells indicating a pro-inflammatory environment were reported. There was also C3 deposition on the RPE basal surface, and increased autofluorescence, likely due to lipofuscin accumulation. In addition, photoreceptor loss occurred with decreased ONL thickness, resulting in an age-dependent reduction in both photopic and scotopic ERGs [23].

### 4.2. Inflammatory Models

#### 4.2.1. CCL2 KO Mouse Model

Mice deficient in the monocyte chemoattractant protein-1 (Ccl-2) gene or the gene coding for its receptor, C-C chemokine receptor-2 (Ccr-2), were reported to exhibit various AMD characteristics. These include drusen deposition, RPE atrophy, RPE hypo-pigmentation, increased autofluorescence and A2E levels, thickened Bruch’s membrane, complement deposition, and photoreceptor atrophy [48]. Its knockout also impaired macrophage recruitment to clear and degrade drusen, hence allowing drusen accumulation. At 6 months, complement deposition were identified in some mice, indicating increased risk of drusen formation. At 9 months, drusen was deposited and Bruch’s membrane thickened. Increased A2E levels were detected at 12 months, while increased autofluorescence occurred at 15 months. At 16 months, RPE atrophy, yielding a phenotype similar to geographic atrophy occurred, together with hypopigmentation and photoreceptor atrophy.

Another study has generated mice KO models lacking both CCL2 and CX3C chemokine receptor 1 (CX3CR1). Cardinal features of age-related macular degeneration were shown to be elicited by subretinal microglia cell accumulation, and this is CX3C chemokine receptor 1 (CX3CR1)-dependent [49]. CCL2/CX3CR1^−/−^ mice were presented with drusen, thickened Bruch’s membrane, increased levels of A2E, increased complement activation, immune cell accumulation, and photoreceptor dysfunction [50]. By 6 weeks, some features of AMD had developed: drusen deposition, photoreceptor dysfunction, and alteration of the RPE layer. At 15 weeks, A2E levels increased. Although the exact time point of development was not specified, mice aged 8–60 weeks old also presented with thickened Bruch’s membrane, local RPE hypopigmentation, and vacuolation, as well as increased complement C3b and C4b activation. Microglia infiltration was also detected in retinal lesions. Notably, choroidal neovascularisation was observed in 15% of the mice [50].

#### 4.2.2. CFH^−/−^ Mouse Model

Variations in several complement genes are known to be significant risk factors for the development of AMD. The most widely studied is complement factor H (CFH), with mutations (e.g., Y402H) that increase the risk for disease progression by up to 7.2-fold in some animals [51,52]. In CFH^−/−^ mice subretinal deposits, increased autofluorescence (suspected to be lipofuscin), complement deposition, photoreceptor dysfunction, and reduced ERGs were observed at 2 years [53]. Additionally, C3 deposition was observed along Bruch’s membrane and the RPE layer, whereas in normal animals, C3 deposition was only present in Bruch’s membrane [54]. Both a-wave and b-wave amplitudes were significantly decreased in scotopic conditions, indicating compromised rod photoreceptor function.

However, this in vivo model also presents several discrepancies when compared with AMD pathogenesis. Photopic conditions did not exhibit any significant decrease in ERG amplitude, suggesting normal cone function. RPE atrophy, commonly associated with AMD, was also not observed in this mouse model [53]. Another study identified delayed retinal development in CFH^−/−^ mice results in decreased retinal thickness. However, the retinal thickness converged with that of wild-type mice at 2 months [55]. Hence, showing that the CFH^−/−^ mouse model may not fully recapitulate dry AMD features.

#### 4.2.3. CFH H402 Mouse Model

CFH H402 mice were also used as a model for dry AMD, given that the variant form of CFH Y402H is an established risk factor for the development of AMD in humans. These mice were fed normal diets until 90 weeks of age, and then switched to a high-cholesterol diet for 8 weeks. They presented with more multinucleated RPE cells, basal laminar deposits, reduced ERGs, and complement activation [56]. These changes were significant when comparing between mice fed with a high-cholesterol diet against those fed normal diets. Notably, plasma cholesterol levels were higher in mice expressing normal CFH protein fed a high-cholesterol diet compared with variant mice fed the same diet. Higher ApoA1 and ApoB100 were observed in the former mouse type. Landowski et al. concluded that phenotypic differences in normal CFH mice and variant mice were due to lipoprotein regulation, not complement activation [56].

#### 4.2.4. C3-Overexpressed Mouse Model

Transgenic C3 overexpression mice present with RPE atrophy, pigmentary changes, complement deposition, photoreceptor loss, and reduced ERGs [57]. These mice were generated via the injection of 6–8-week-old C57B16/J male mice with recombinant adenovirus-expressing murine complement component 3, using a transscleral/trans choroidal approach. Eight days post-intervention, photoreceptor outer segment loss was noted and likely attributed to the deposition of membrane attack complexes on it. Prior studies have reported that the formation of membrane attack complexes on RPE cells leads to cell lysis [58,59]. At 9 days post-intervention, RPE loss and hypopigmentation were also observed. In addition, retinal detachment occurred. A notable mention of this model would be that it shows features of neovascularisation, a key phenotype of wet AMD. This was demonstrated by fluorescein leakage during fluorescein angiography at 8 days post-intervention due to more permeable blood vessels. Furthermore, no drusen deposition was observed. Therefore, this study presents an in vivo AMD model for both dry and wet AMD [57].

#### 4.2.5. CFH^−/−^ C3^−/−^ KO Mice

CFH^−/−^ C3^−/−^ mice were also investigated at the age of 12 months. They presented with thickened Bruch’s membrane, sub-RPE deposits, RPE loss, pigmentary changes, photoreceptor loss, and reduced ERGs [54]. Photoreceptor atrophy was noted with thinning of the outer nuclear layer (ONL). Interestingly, a reduced macrophage number, but a chronic proinflammatory state was observed, based on TNF-α and calcitonin levels in the photoreceptor layer as markers for inflammation [54]. This substantiates the inflammaging as a theory of aging.

#### 4.2.6. ApoE Mouse Model

ApoE gene variants have been correlated with AMD incidence. In humans, ApoE exists as three allelic variants: E2, E3, and E4. In the past decade, it has been established that ApoE4 is protective in AMD, whereas ApoE2 plays a role in instigating AMD [59]. Hence, mouse models expressing different ApoE polymorphisms have been utilised to investigate their effect on AMD development. Interestingly, a mouse model expressing human ApoE4 identified the greatest severity of AMD phenotypes, despite its reported protective effects. This mouse model presented with drusen and sub-retinal deposits, RPE atrophy, RPE pigmentary changes, and a thickened Bruch’s membrane [60]. These findings were present in mice aged 65–127 weeks old, who were fed a high-fat, cholesterol-rich diet. All findings were more severe in human ApoE4-expressing mice compared with human ApoE2-expressing mice, except for RPE atrophy, which was only observed in ApoE4 mice. In addition, some choroidal neovascularisation was also observed.

### 4.3. Light-Induced Oxidative Stress Models

Light-induced retinal degeneration in C57BL/6 mice with the leucine-to-methionine substitution at the 450th residue (L450M) of RPE65 causes subretinal debris, RPE atrophy, complement deposition, and immune cell accumulation. L450M increases the susceptibility to light-induced damage as rhodopsin, a GPCR found in rods, regenerates quickly and enables additional light absorption. The mice were treated with an intraperitoneal injection of 2 mg fluorescein. This agent is commonly used to visualise blood vessels in the eye [61,62]. However, one study reported that exposing the eye to both fluorescein and visible light may induce light damage to highly susceptible retinas [63]. Ten minutes later, the mice were exposed to a 54 K lux light for four minutes. Subretinal debris was observed 21 days after the intervention, while RPE atrophy was observable on both day 10 and day 21. C3d factor deposition was noted 3 days after the intervention. Mice treated with 125 K lux light exposure for 30 min exhibited subretinal inflammation as early as 2 days after light injury [64].

### 4.4. Antioxidant Gene Knockout Models

The knockdown of genes involved in the antioxidant pathway disrupts the careful balance in maintaining the oxidative load. One such example is the knockdown of the superoxide dismutase (SOD) family, which plays a critical role in catalysing superoxide radicals to H_2_O_2_ and O_2_. It has three isoforms: SOD1, SOD2 [65], and SOD3 [66].

SOD1 knockout mice presents with drusen deposits, RPE degeneration, photoreceptor atrophy, thickened Bruch’s membrane, and reduced ERGs. At 7 months, drusen-like retinal deposits which were hyperfluorescent during fluorescein angiography were observed. This was further confirmed at 12 months, where markers of drusen were detected via immunohistochemistry. In addition, RPE degeneration and photoreceptor loss were observed in some mice at 12 months, alongside Bruch’s membrane thickened by the vacuolation of RPE cells [67]. Decreased rod and on-bipolar cell activity were shown, correlating with photoreceptor loss. Additionally, the retina layer was found to decrease in thickness as early as 10 weeks, with significant decreases at 50 weeks. Damaged mitochondria in ONL were also observed.

### 4.5. Other Mouse Models

One of the most promising models recently published involves CLIC4 mice, in a chloride intracellular channel 4 (CLIC4) deletion model, which shows several features such as a thickened Bruch’s membrane, drusen-like deposits, RPE cell death, fundus pigmentary changes, complement deposition, immune cell accumulation, photoreceptor atrophy, and reduced ERGs. CLIC4 serves a variety of functions that help maintain RPE homeostasis. These include signal transduction, cell adhesion and migration, cell death, and gene regulation [68].

At 3 months, white-yellowish fundus lesions begin to appear, and the basal RPE deposition of C3 and its activation products (C3b/iC3b/C3c) are observed. At 6 months, there is also increased macrophage infiltration, which peaks between 6 and 9 months. The Bruch’s membrane was also thickened, corresponding to lipoprotein aggregates. Photoreceptor atrophy was reported at only 12 months with the thinning of the ONL layer. At 12 months, drusen-like deposits were observed, alongside RPE cell loss, which led to patches of hyper/hypopigmentation of the fundus. These patches were attributed to massive RPE cell death. In addition, they also revealed underlying choroidal vessels, suggesting potential choroidal neovascularisation. Notably, CLIC4 deletions altered the expression of 45 AMD risk genes involved in numerous functions, including lipid metabolism, innate immunity, oxidative stress, and angiogenesis, amongst others. The expression of cytokines/cytokine receptors related to AMD such as CCl2, Ccr2, and Cx3cr1 were also modified [68].

Another important model of AMD utilises the nutritional deficiency of B vitamins (B12 and folic acid, which are involved in the methionine or one-carbon pathways) using mice with elevated homocysteine (CBS mice). This mouse was reported as a model of AMD (both dry and wet). Elevated homocysteine showed AMD features both in vivo and in vitro where “Hyperhomocysteinemia disrupted RPE structure and function with features of AMD” [69]. This mouse showed the activation of oxidative stress, ER stress [70], epigenetic modification [71], inflammation [72], and induced a metabolic switch in the mitochondria, in which RPE cells predominantly produced energy through the high rate of glycolysis (Warburg effect) [73].

### 4.6. Summary for AMD Disease Modelling

Although the results of many animal studies performed in models of AMD are encouraging, it is important to remember that no model has fully replicated the human disease so far. All of the animal models used to study AMD have limitations, and the results should therefore be interpreted with caution, particularly concerning how they apply to the human condition. These models differ with regard to the mechanisms of RPE cell loss, the importance of inflammation, and the recruitment of other cells during the degenerative process. This brings up the question of the model’s physiological relevance to AMD. Hence, better animal models of AMD are an important goal for future research. We have made a comparative table for some of the important animal models with their features available for AMD research (see Table 1).

## 5. Clinical Trials Targeting Dry AMD

Multiple early-phase clinical trials have shown significant promise, creating high expectations from the associated expansion phases that are ongoing or anticipated in the near future [1]. There have also been trials that have failed to meet primary and secondary endpoints, despite promising results in animal models and ex vivo. The main difficulties have revolved around the effectiveness of one drug for patients in different stages, various methods of drug delivery, the lack of efficacy in reducing the rate of atrophy progression as compared with control eyes, and safety challenges. Due to its complexity, various therapeutic targets have been considered and include visual cycle modulation, neuroprotection, cell-based therapy, inflammation suppression, and complement inhibition. Nevertheless, given the increasing global burden of dry AMD, continued rapid development is expected, with the number of clinical trials increasing into the foreseeable future. In this section, we will examine recent translational research for dry AMD.

### 5.1. Nutritional Supplementation and Antioxidant Therapy

Oxidative damage to the retina from various sources, such as smoking, UV light exposure, and oxidative stress, have been strongly linked with AMD. Hence, treatments that reduce the accumulation of ROS may be a potential therapeutic intervention. The age-related eye disease study (AREDS) was a double-blind, randomised, multi-centre trial (*n* = 3640; NCT00000145) that was designed to determine the protective effects of antioxidant supplementation in patients with AMD [74]. This study recruited participants with varying degrees of AMD severity, ranging from extensive small drusen in retina to people with vision loss in one eye due to AMD. They were given either placebo, high doses of antioxidants (beta carotene, vitamin E, and vitamin C), zinc, or both.

Interestingly, combinatorial treatment of antioxidants and zinc worked best only for patients with intermediate AMD severity and beyond. A loss of visual acuity was significantly attenuated in this group through the combination treatment (*p* = 0.008). It should also be mentioned that the administration of high doses of zinc or antioxidants generally did not induce an adverse effect. In summary, this clinical study demonstrated the potential of nutritional supplementation at slowing down disease progression for intermediate and late AMD (wet and dry). However, the efficacy of this treatment should be further studied in the context of wet AMD, because it did not address its efficacy on participants that already had neovascular AMD in both eyes.

The AREDS2 clinical trial was also extended to study the effect on cognitive function when lutein/zeaxanthin is additionally supplemented. NCT00345176 was a controlled randomised clinical trial, where the participants were evaluated to have high risk of progression to late AMD [75]. They have found that there is no significant effect on the prevention of cognitive decline in AMD patients. However, there are several caveats to this study. For example, a baseline for cognitive testing was not established; hence, it was unlikely to determine how the ARED2 supplement could have originally affected cognitive function. Therefore, it may be too early to rule out the possibility that the addition of lutein/zeaxanthin has no positive effect on cognitive function.

### 5.2. Visual Cycle Modulation

Visual cycle modulators are oral medications that target enzymes in the visual cycle. In phototransduction, photoreceptors exert a high metabolic demand, which results in the increased production of metabolic waste products. The accumulation of these by-products may lead to increased inflammation, which is implicated in the development of GA. Modulating the visual cycle may mitigate this process and reduce inflammation and GA. A drug that can be delivered orally is especially attractive; however, a consequence of modulating the visual cycle is that dark adaptation and low-light vision are often adversely affected.

ALK-001 is a modified form of vitamin A that replaces natural vitamin A in the body. Modified vitamin A forms the toxic vitamin A dimer more slowly, which is postulated to slow the accumulation of toxic end products, and therefore slow the development and/or progression of AMD. However, this effect may also impact the visual cycle by depleting vitamin A and causing delayed dark adaptation. It is currently being studied in a Phase III clinical trial for GA [76].

### 5.3. Neuroprotection

Neuroprotection has been investigated as a possible solution for the problem of progressive cellular damage and eventual cell loss that occurs in atrophic AMD. Pharmacologic agents with cyto- and neuroprotective properties may help protect at-risk neuro-retinal tissue by increasing its resilience and resistance to cellular injury, thereby providing a defence against GA progression. Thus far, brimonidine tartrate (Allergan) is the only agent that has shown possible neuroprotective properties that might be beneficial in GA. Brimonidine is an alpha-2 adrenergic agonist that is an established topical ophthalmic intraocular-pressure-lowering agent. However, studies in animal models with systemic administration have demonstrated that it also has neuroprotective properties, although clinical trials have yet to confirm similar efficacy in humans [77,78].

### 5.4. Mitochondria-Based Therapies

Based on the central role of mitochondria in AMD, strategies targeting mitochondrial homeostasis have great potential [79]. Interestingly, subcutaneous treatment with Elamipretide (Stealth Biotherapeutics, Needham, MA, USA), a tetrapeptide SS-31 drug which targets mitochondrial cardiolipin, is currently under development for dry AMD, reversed RPE morphological changes, sub-RPE deposits, and visual dysfunction. Elamipretide is a cell-permeable peptide delivered via a 40 mg subcutaneous injection that targets mitochondrial dysfunction [80,81]. SS-31 is currently undergoing a Phase II randomised, double-blind, placebo-controlled clinical trial to evaluate its safety, efficacy, and pharmacokinetics in subjects with dry AMD (NCT03891875vi) [82,83,84].

Risuteganib (Allegro Ophthalmics, San Juan Capistrano, CA, USA), is another promising drug under investigation for dry AMD, as well as other retinal indications such as diabetic macular edema [85]. Risuteganib, an integrin antagonist, is reported to have multiple mechanisms of action, including mitochondrial protection. Risuteganib has been shown to prevent mitochondrial injury in cultured RPE cells exposed to hydroquinone, suggesting that mitochondrial protection is efficacious in an in vitro model of AMD [86]. Additionally, risuteganib was found to reduce mitochondrial ROS and improve mitochondrial bioenergetics in cultured RPE [87].

On the basis of these preclinical data, both Elamipretide and risuteganib have advanced to human trials and have completed early-stage clinical studies showing promising signs of efficacy in patients with dry AMD [88].

### 5.5. Metformin

Metformin, a common anti-diabetic drug, has been shown to have protective outcomes in multiple age-associated diseases and may have the potential to protect against the development of AMD [89]. In a case-controlled study of a population-based sample of patients from a nationwide health insurance claims database, it was shown that metformin was associated with preventing the development of AMD. This association was dose-dependent, with the greatest benefit at low to moderate doses. When looking only at patients with diabetes, a preservation of the dose-dependent decrease in the chances of patients developing AMD was found. The significant odds ratio of 0.58 shows that diabetic patients taking metformin compared with a DPP4 inhibitor (antidiabetic drug) had a significantly decreased risk of developing AMD [89]. However, metformin does not appear to be protective in patients with diabetes and coexisting diabetic retinopathy [90]. Future studies will be important to further validate and confirm this finding, in addition to determining the molecular mechanisms involved and which pathogenic pathways of AMD are affected by metformin.

### 5.6. Photo Biomodulation

In addition to chemical modulators of mitochondria, far-red to near-IR light (590–850 nm) therapy, referred to as photo biomodulation (PBM), has also demonstrated beneficial effects in multiple retinal degenerative models and patients with AMD [91]. PBM targets mitochondrial cytochrome C oxidase, a protein that modulates the transfer of electrons between the electron transport chain complexes, increasing the mitochondrial membrane potential and ATP synthesis. In aged mice and an AMD mouse model with a genetic disruption of complement factor H, PBM significantly increased retinal mitochondrial function and reduced signs of inflammation [92,93]. These studies provide insight into the molecular basis for the beneficial effects of PBM. Current evidence for the efficacy of PBM in AMD is poor, but its safety profile and proposed mechanisms of action motivate further research as a novel therapy for AMD.

LIGHTSITE III, a prospective, double-blind, randomised, multi-centre clinical trial, was conducted at ten leading US retinal centres [94]. This study aimed to treat dry AMD subjects with PBM every 4 months for a duration of 24 months. The trial results demonstrated statistically significant improvements in the prespecified primary endpoint in visual acuity at 13 months in the PBM treatment group over the sham treatment group (*p* = 0.02). The small number of patients (*n* = 100) used in this study is a limitation for power calculation, and thus, it is not sufficient for drawing robust conclusions.

The abovementioned therapies provide the clinical landscape for upcoming new treatments for AMD. These data show promise in developing alternative treatments for AMD and have demonstrated that it is possible to target and modulate several pathways. Therefore, they display the potential for future studies to develop more innovative solutions to slow down and hopefully stop the progression of AMD. A compilation of the current and past clinical trials for AMD research is shown in Table 2.

## 6. Conclusions

Although multiple targets such as complement inhibition, neuroprotection, and anti-inflammatory factors have been investigated for the treatment of AMD, these findings have currently not been translated into clinical treatment options. Translational medicine still has to pave the way in identifying the target populations that could benefit from such therapy. These treatment failures can be justified by the concept of “the point of no return” in the disease cascade process, which has led to irreversible cell loss (i.e., RPE and photoreceptors). The current clinical approach in the management of dry AMD is focused on dietary supplementation to prevent conversion to late stages of the disease without clear visual benefit [95].

Although recent advances highlight the importance of mitochondrial homeostasis in disease mechanisms, several outstanding questions and key issues remain to be investigated, including the potential mechanisms involved in the higher bioenergetics demand of the macula and metabolic uncoupling of the retinal ecosystem with AMD. Therapeutic strategies targeted towards maintaining mitochondrial homeostasis (structural, functional, and genomic) have the potential to prevent the development and progression of dry AMD. 

## Figures and Tables

**Table 1 ijms-24-08763-t001:** Compilation of animal models available currently for AMD research.

	Thickened BM	Sub-Retinal Deposits/Drusen	RPE Hypoplasia/ Atrophy	Increased Autofluorescence	Complement Deposition	Photoreceptor Atrophy/ Dysfunction	Reduced Electroretinograms	Reference
**RB1CC1 conditional KO mice**		×4 mo	×4 mo	×8 mo	×8 mo			[23]
**NRF2^−/−^**	×12 mo	×8–11mo: hard drusen, 11–18 mo: soft drusen	×12 mo	×12 mo	×12 mo		×12 mo	[43]
**NRF2^−/−^ PGC1a^−/−^**	×12 mo	×12 mo	×12mo			×12 mo	×12 mo (only b-wave statistical significance)	[45]
**Cx3cr1^−/−^**		×18 mo	x		×18 mo	×4 mo		[49]
**Ccl2/Cx3cr1^−/−^**	×8–60 wk	×6 wk	x		x	×6 wk		[50]
**Cfh^−/−^ mice**	thinning	×2 years		×2 years	×2 years	×2 years	×2 years	[53]
**Cfh^−/−^ C3^−/−^**		×12 mo	×12 mo			×12 mo	×12 mo	[54]
**APOE e2/e4 transgenic mice +/− high fat**	×* 65–127 wk	×* 65–127 wk	×** 65–127 wk			×**		[60]
**Light-induced model**		×21 days after exposure to 54 K lux	×RPE atrophy both 10 and 21 days		×3 days after exposure to 4 min 54 K lu×			[64]
**SOD2 KO**	×4 m after injection of AAV		×2 m after injection of AAV	×4 m after injection of AAV		×2 m after injection of AAV	×2–6 months	[65]
**SOD1 KO**	×12 mo	×7 mo	×12 mo	×7 mo		×12 mo	×40 wk	[67]
**CLIC4 mice**	×12 mo	×12 mo	×12 mo	Reduced 12 mo	×3 mo	×12 mo	x; lower a-wave amplitude rod 9 mo, cone 12 mo	[68]

* APOEe4 > APOEe2, when on fat diet, ** only in APOEe4 mice.

**Table 2 ijms-24-08763-t002:** Compilation of ongoing and past clinical trials in AMD.

Target	Drug (Sponsor)	Administration	Phase	Design	Sample Size (n=)	Status	First Posted	Trial Number	Clinical Outcomes	Reference
Oxidative Pathway	Lutein/zeaxanthin and DHA/EPA (NEI) + AREDS formulation, AREDS2	Oral	III	Added to AREDS formulation	4023	Completed	2006	NCT00345176	Lack of efficacy	[74]
Visual Cycle modulator (Retinaldehyde)	ALK-001 (Alkeus Pharmaceuticals)	Oral	III	Daily vs. sham	300	Ongoing	2019	NCT03845582	-	[76]
Reduction of toxic by-products	GAL-101 (Gemini Thera-peutics Inc.)	Intravitreal	I	Single Ascending dose	12	Completed	2020	NCT04246866	Safe and well tolerated	[82]
Cardiolipin-protective compound	Elamipretide (Stealth BioTherapeutics)	Subcutaneous	I	One dose daily in high risk drusen vs. noncentral GA vs. no intervention	49	Completed	2016	NCT02848313	Safe and well tolerated, improvement in low luminance and BCVA	[83]
Mitochondrial Enhancers	Elamipretide (Stealth BioTherapeutics)	Subcutaneous	II	Daily vs. sham	176	Completed	2019	NCT03891875	-	[84]
Integrin heterodimers	Risuteganib (Allegro Ophthalmics)	Intravitreal	II	Luminate vs. sham	42	Completed	2018	NCT03626636	Primary endpoint met in 48% of patients with treatment vs. 7.1% in sham, statistically significant	[87]
Mitochondrial Enhancers	Metformin	Oral	II	Case control study	95	Ongoing	2022	NCT02684578	N/A	[90]
LIGHTSITE III study	Valeda Light Delivery System	Interventional	N/A	Multiple	96	ongoing	2019	NCT04065490	none	[94]

## Data Availability

Not applicable.

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
