# Peer review of "Dysfunctional Autophagy, Proteostasis, and Mitochondria as a Prelude to Age-Related Macular Degeneration"

_ijms, 2023, doi:10.3390/ijms24108763_

Round 1
Reviewer 2 Report
Raji Rajesh Lenin an coworkers have undertaken tremendous efforts to present an overview over “Mitochondrial dysfunction in age-related macular de-1 generation”. This manuscript contains a lot of interesting information, but reading it is challenging and confusing, because the many exciting findings from the multiple cited in vitro and in vivo studies are listed, but not set into a context which leaves the reader with a large number of stones in a puzzle where obviously also the authors do not see the picture and where the single information is to be placed. Adding thereto, reading is demanding based on a limited linguistic quality which should definitely be improved. It seems, that the more confusing it was for the authors to put the information from the multiple studies into a context the more they struggled with sentence construction and the creation of a fact-based message. Further important points:
After each chapter, not only after chapter 4, a short concluding statement (Taken together, these findings indicate… or similar) should be displayed which provides the frame or context picture, into which the aforementioned studies are to be set.
Multiple parallel-acting mechanisms and compensatory factors may be or have been established at several points. These can be overcome in vitro, but not so easily in vivo, and namely not in knock out models. This is a major reason why current knowledge is so difficult to set into a context. A corresponding chapter deserves to be created at least in the animal models chapter to help the reader to understand the heterogenous and partially contradictory outcomes. One typical example is found in line 263: This sentence should be reworded to “… that both genes relevantly contribute to the control of stress regulation (in an additive or more than cumulative fashion?).
The differentiation of inflammatory and complement pathway models may have its justification, but the authors do hardly report any inflammatory model not linked to complement factors (i.e. ref 52). Would fuse the chapters or, which I personally would prefer, because it comes also later into focus, extend to information regarding the role of inflammasomes, cellular infiltration of degenerative lesions leading to the concept of inflammaging.
Minor points:
Introduction, line 61: which are the limited if at all existing treatment options beyond belief and hope?
The last paragraph of the intro is important, therefore re-wording is needed to understand the authors’ intention beyond a listing of topics to be tackled. For example, what is “mitochondrial dysfunction in A-D”?
Bruch’s membrane thickening is key to advanced dry AMD. Talking about it therefore deserves an elaboration, how this develops according to current concepts (line 90)
The authors are happy with listing findings with the word “also” which is mis-placed and uninformative at many places, starting at line 95.
Abbreviations need always to be explained, even if they are well-known in the community, such as in lines 106, 414 and 617.
P3, par 2 and 3 need to be rewritten.
Line 110: phenotype of what?
Line 122: What is an “effective” in vitro and what a meaningful in vivo model? At the begin of the “in vitro” and “in vivo” chapters, the authors should provide a short statement of the minimal conditions that are to be met to make a model meaningful. Effective may not be the ideal term here.
What is meant with “This” in line 182? For example: “The lack of generally accepted standard protocols challenges the comparison of different studies’ outcomes and highlights…”?
Line 253: “necessary”? Think it contributes beyond others in a multifactorial disease process to the regulation…
Lines 346-9: Message unclear? Pls re-word. What means “which does not model after dry AMD as well”?
Line 372: What are membrane attack complexes?
Lines 400-2: re-word, contradictory and unclear.
Lines 600-5: Pls re-word.
Line 607: Compared to diabetics without metformin?
Chapter 5.6: Lightsite has made lot of rumours and is currently aggressively marketed. Please report the limitations of Lightsite studies (i.e. the power, which is not given with 143 patients in Lightsite III to create a robust evidence.
Line 629: “… yielded positive results” is not accurate. These findings have currently not been translated into clinical treatment options. Translational medicine has still to pave the way namely to identify the target populations that could benefit from such therapy.
Concluision, 2nd par, lines 635-41: The pertaining information has NOT been reported above. This par currently refers to nothing. Set into context after having provided the corresponding information.
References need to be checked for appropriateness and completeness. Beyond others, refs in lines 90, 128, for example seem wrong, refs 50 and 71 incorrect, refs 88-96 not discussed in the text.
Tables 1 and 2 not embedded in the text.
In conclusion, lots of interesting information, but in the current state of presentation not mor useful than reading the single papers, as long as a clear contextual frame is not provided.
Reviewer 3 Report
A review manuscript entitled “Mitochondrial dysfunction in age-related macular degeneration” is authored by Raji Rajesh Lenin. et al.
A comprehensive review aiming to summarize the up-to-date published evidence of mitochondrial dysfunction in dry AMD, in-vitro and in-vivo disease models relevant to assess mitochondrial dysfunction in AMD, and their effectiveness in drug screening and current clinical trials targeting mitochondrial dysfunction for AMD therapeutics.
Well-written inclusive review that covered very important aspects in the pathogenesis of AMD especially the dry type of AMD. Mitochondrial dysfunction is a well-accepted mechanism in Dry AMD, that leads to activation of many other pathways that accelerate the disease progression. However, there are some issues that need to be addressed to improve the quality of the manuscript before publication.
1-The part from 176-184 has too many details, please summarize starting with ‘For studies using TBHP,….
2-The part 177-202, needs to be rewritten in a more constructive way
3-The review showed the different In -vivo models for investigating dry AMD phenotypes, however, it didn’t include a very important model, of nutritional deficiency of vitamins B (B12 and folic acid that are involved in the methionine or one carbon pathways (mice with elevated homocysteine, CBS mice). This mouse was reported as a model of AMD (both dry and wet). Elevated homocysteine showed AMD featured both in vivo and in vitro “Hyperhomocysteinemia disrupts retinal pigment epithelial structure and function with features of age-related macular degeneration” PMID: 26885895. This mouse showed ERG a- and b-wave and the light peak component changes (PMID: 22197750), activation of oxidative stress (PMID: 28931831), ER stress (PMID: 25580465), Epigenetic modification (PMID: 29560091), inflammation (PMID: 32138265), inducing a metabolic switch in the mitochondria, in which RPE cells predominantly produce energy by the high rate of glycolysis (Warburg effect) PMID: 36674587, activation of N-methyl-D-aspartate receptor (NMDAR) with suggested the FDA approved NMDAR blocker Memantine as a therapeutic target for AMD (PMID: 345022660)
4-4.1. NRF2/PGC - 1α and RB1CC1 deficient mice models, please subdivide the latter into
4.1a. NRF2
4.1b. PGC
4.1c. NRF 2/PGC -1α double KO
4.1.d. RB1CC1
5-Please rephrase “Macrophage infiltration was also noted, after staining for the CD68 marker. At 8 months, the severity increased and Iba -1 positive cells indicating activating microglial cells were noted”
6-This part is confusing (line 350) “CFH H402 mice were also used to model after AMD” please change to CFH H402 mice were also used as a model for AMD.
7-Line 369” These mice were generated via injection of 6- to 8-week old” please change to 6-8 weeks old or 6 to 8 weeks old.
8-Line 377-379’ However, features of neovascularization were noted, with fluorescein leakage during fluorescein angiography at 8 days’ post-intervention. This was due to more permeable blood vessels” This means that this isn’t a model of dry AMD only (as the title of the animal models indicated dry only) but both dry/wet types of AMD.
9-Part ‘4.5. Oxidative Stress Models –Light induced, antioxidant gene KO” needs to be rephrased. Please make sure of this sentence as IP injection of fluorescein is used to visualize the blood vessels during fluorescein angiography ‘The mice were treated with an intraperitoneal injection of 2mg fluorescein, which further increases the susceptibility of the retina to light injury. 8 minutes later, the mice were exposed to 54K lux light for 4 minutes”
As shown in the same part below These mice were backcrossed to C57BL/6 and wild -type mice. At 435 7 months, drusen -like retinal deposits which were hyperfluorescent during fluorescein angiography were observed
10-Please elaborate more details about the nutrition used in this study and what were the results of the study “. Age -related eye disease study (AREDS) supplements AREDS was a double-masked, randomised and multicentre trial (n=3640; NCT00000145) that was designed 527 to determine the protective effects of antioxidant supplementation in patients with AMD”
Round 2
Reviewer 2 Report
The authors have improved their manuscript, though the linguistic quality is still flawed. As one example, "A cardinal feature of age -related macular degeneration being subretinal microglia cell accumulation is CX3C chemokine receptor 1 (CX3CR1) -dependent ( 61)."
If the authors would invest a little for improving the linguistic quality of their paper and, as suggested, shorten the manuscript to the information that has been set into a robust pathophysiological context, readability and the chance for citations would clearly gain.
Round 3
Reviewer 2 Report
The manuscript enjoyed remarkable improvement. Minor changes might apply. These include:
The 1st sentence in the abstract should be moved down to before “aged mitochondria”
Pls have a spell and grammatical check done after accepting changes.
Re-word last sentence on p7
L 473: replace individuals by animals
Line 798: re-word, afte shortening out of context
Line 845: Ref 108 wrong
Citations 39 and 40 identical
Ref 89 needs updating incl clinicaltrials.gov identifyer
Ref 97 is incomplete, pls correct
